# Leveraging Probabilistic Segmentation Models for Improved Glaucoma Diagnosis: A Clinical Pipeline Approach

**Anna M. Wundram**[*1]                     ANNA.WUNDRAM@STUDENT.UNI-TUEBINGEN.DE
**Paul Fischer**[*1]                                    PAUL.FISCHER@UNI-TUEBINGEN.DE
**Stephan Wunderlich**[1,2]                              STEPHAN.WUNDERLICH@TUM.DE
**Hanna Faber**[3,4,5]                                                  H.FABER@UKE.DE
**Lisa M. Koch**[6,7]                                            LISA.KOCH@UNIBE.CH
**Philipp Berens**[1,6]                                 PHILIPP.BERENS@UNI-TUEBINGEN.DE
**Christian F. Baumgartner**[1,8]                    CHRISTIAN.BAUMGARTNER@UNILU.CH

[1] *Cluster of Excellence – Machine Learning for Science, University of Tübingen, Germany*

[2] *Ludwig-Maximilians-University of Munich, Germany*

[3] *University Clinic Hamburg, Germany*

[4] *University Eye Clinic Tübingen, Germany*

[5] *Moorfields Eye Hospital, London, UK*

[6] *Hertie Institute for AI in Brain Health, University of Tübingen, Germany*

[7] *Department of Diabetes, Endocrinology, Nutritional Medicine and Metabolism UDEM, Inselspital, Bern University Hospital, University of Bern, Switzerland*

[8] *Faculty of Health Sciences and Medicine, University of Lucerne, Switzerland*

**Editors:** Accepted for publication at MIDL 2024

## Abstract

The accurate segmentation of the optic cup and disc in fundus images is essential for diagnostic processes such as glaucoma detection. The inherent ambiguity in locating these structures often poses a significant challenge, leading to potential misdiagnosis. To model such ambiguities, numerous probabilistic segmentation models have been proposed. In this paper, we investigate the integration of these probabilistic segmentation models into a multistage pipeline closely resembling clinical practice. Our findings indicate that leveraging the uncertainties provided by these models substantially enhances the quality of glaucoma diagnosis compared to relying on a single segmentation only.

## 1. Introduction

Glaucoma is a chronic eye disease in which nerve fibers gradually degenerate, leading to damage in the optic nerve head. It is the second leading cause of blindness worldwide affecting one in ten people over the age of eighty (Sevastopolsky, 2017). Diagnosis involves the exact localization of the optic disc and cup in the fundus image. Their shape, size, and relationship to each other are crucial disease markers. Therefore, automated segmentation of the cup and disc is an important step for computer-aided diagnosis frameworks. However, delineation of those structures, in particular the optic cup, is highly challenging and subject to large uncertainties with large disagreements even among experts.

---

* Contributed equally

In this paper, we demonstrate that probabilistic segmentation techniques are effective at capturing uncertainties in the segmentation of the optic disc and cup. Going further, we propose a method to propagate this uncertainty through a multi-stage diagnostic pipeline. Specifically, we propose a method for incorporating the uncertainty arising in an upstream step in the final downstream task, and show that it substantially improves classification performance with respect to a deterministic baseline[1].

Several models have been proposed for optic cup and disc segmentation over the past years (Sevastopolsky et al., 2019; Tulsani et al., 2021; Rasheed et al., 2023). However, these methods do not take into account the inherent uncertainties of this problem. A small number of works address expert disagreements. Edupuganti et al. (2018) incorporate multi-annotator information into learning the segmentation by weighing the loss for pixels depending on the agreement or disagreement of the annotators. Cheng et al. (2020) approximate the joint distribution of the input fundus image and the ground truth segmentation to regularize a U-Net segmentation network. However, estimation of the segmentation uncertainty in optic cup and disc segmentation remains unexplored.

Most recent automated glaucoma diagnosis frameworks approach the problem with black box solutions (Mirzania et al., 2021; Singh et al., 2022; Fan et al., 2023; De Vente et al., 2023). While reaching high performance, end-to-end black box models may often not be clinically desirable. In practice, systems that model the individual steps of a diagnostic pipeline are more interpretable, easier to debug, and more likely to find clinical acceptance. Moreover, data shifts can be addressed by retraining individual components rather than the whole method. Indeed, clinical pipeline approaches have shown great promise in similar areas (De Fauw et al., 2018). A number of works propose to first segment the cup and disc and then extract the vertical or area *cup-to-disc ratio* (CDR) as a diagnostic marker (Al-Bander et al., 2018; Sevastopolsky et al., 2019; Bi et al., 2019; Jiang et al., 2019; Bian et al., 2020; Neto et al., 2022; Zhang et al., 2023). The vertical CDR measures the ratio between the diameter of the cup and the disc along a vertical line through the center. The area CDR measures the ratio of pixels belonging to the cup and disc respectively. However, clinical literature shows that the *rim thickness curve* (RTC) (i.e. the distance between the disc and the cup for every point) may be a more informative measure (Spaeth et al., 2002; Kumar et al., 2019). In this work, we follow these insights and propose a clinically inspired multi-stage pipeline based on initial probabilistic segmentation of the cup and disc, automated RTC extraction, followed by glaucoma classification (see Fig. 1).

Uncertainty estimation is typically studied for individual deep learning models, and has been shown to yield improved performance and robustness on individual tasks (Abdar et al., 2023; Schmidt et al., 2023). However, when clinical workflows consist of tasks that are arranged in a cascading sequence – as is the case here – it becomes crucial to understand how uncertainty in certain stages impacts subsequent tasks. Few studies have focused on uncertainty propagation in clinical pipelines. Eaton-Rosen et al. (2018) propagate tumor segmentation uncertainty to tumor volume measurement using the variance sum law. Mehta et al. (2019, 2021) propose to append variance maps obtained using MC Dropout (Kendall et al., 2016) in a segmentation stage as an additional input channel to a subsequent tumor detection network. To our knowledge, these are the only studies showing that incorporat-

---

1. The code is available at https://github.com/annawundram/glaucoma-diagnosis-pipeline

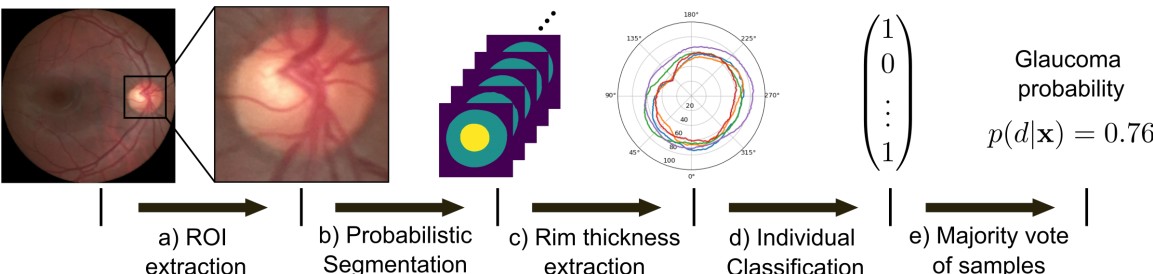

Figure 1: **Proposed pipeline.** a) Automatic ROI extraction from the full fundus images. b) Sampling of possible segmentations using probabilistic segmentation. c) Extraction of rim thickness curves (RTCs) for each segmentation sample. d) Classification of each RTC. e) Marginalization over possible segmentations produces the final glaucoma probability.

ing uncertainty can enhance downstream task performance. However, relying on variance maps instead of the full distribution limits the downstream task to neural networks, as the variance must be added as a channel. Recently, Feiner et al. (2023) and Fischer et al. (2023) proposed two related approaches for propagating uncertainty arising in MR reconstruction to subsequent classification, regression, or segmentation tasks. Both these approaches directly propagate samples from the conditional distribution through the pipeline rather than relying on the variance. Similar, to those works, we propose a sampling-based uncertainty propagation approach. However, we go one step further by using uncertainty to *enhance* downstream performance. This is achieved by marginalizing over possible outcomes in the highly uncertain segmentation stage.

In summary, our contributions are as follows:

1. An interpretable pipeline for glaucoma diagnosis which incorporates clinical knowledge and closely mirrors clinicians' reasoning.

2. The first application of segmentation uncertainty quantification techniques to cup-and-disc segmentation in fundus images with a comparative analysis of four widely used techniques.

3. A sampling-based approach for propagating uncertainty through a multi-stage pipeline as well as a strategy for leveraging this approach to enhance downstream performance.

4. Introduction of RTC as a clinically motivated alternative to CDR in computer-aided glaucoma diagnosis pipelines and empirical verification of its superior performance.

## 2. Methods

The proposed pipeline consists of five steps as illustrated in Fig. 1. We deterministically extract the region of interest (ROI) containing the cup and disc to obtain the close-up images $\mathbf{x}$. For the highly uncertain step of optic disc and cup segmentation, we use probabilistic segmentation to approximate the distribution $p(\mathbf{s}|\mathbf{x})$ of the segmentations $\mathbf{s}$, and we produce

segmentation samples $\mathbf{s}_i$. The samples (which capture the uncertainty) are propagated through a deterministic rim thickness extraction function $g : \mathbf{s}_i \mapsto \mathbf{r}_i$ producing RTCs $\mathbf{r}_i$. Following this, a deterministic classification function $f : \mathbf{r}_i \mapsto d_i$ takes the RTCs as input and produces a predicted diagnosis $d_i \in \{0, 1\}$ for each $\mathbf{r}_i$. Lastly, we marginalize over the samples to obtain a final probability $p(d|\mathbf{x})$ of the image being "glaucoma suspect". All of these steps will be introduced in detail below.

## 2.1. Automated Region of Interest Detection

The region of interest (ROI) for glaucoma diagnosis only includes a small area of the fundus containing the optic nerve head, in particular the optic disc and cup (see Fig. 1a).

Prior work has shown that a two-step segmentation approach consisting of ROI extraction, followed by segmentation can improve segmentation results (Kim et al., 2019; Liu et al., 2021). Following these works we use a U-Net for cup and disc segmentation on full-view fundus images. Next, a padded quadratic bounding box is placed around the segmentations, and the resulting ROI is cropped and resized to $320 \times 320$ pixels. All experiments in this paper were performed with ROI images $\mathbf{x}$ obtained in this manner.

We use the improved U-Net first employed by Kohl et al. (2018) which operates on 7 rather than 5 resolution levels and uses bilinear upsampling instead of transposed convolutions for all experiments, to ensure consistency with the Prob. U-Net (Kohl et al., 2018) and PHiSeg (Baumgartner et al., 2019) baselines described in the next section.

## 2.2. Probabilistic Optic Cup and Disc Segmentation

The cup and disc segmentation step is characterized by large uncertainties and variability even among human experts. Therefore, segmentations resulting from a deterministic approach, such as a U-Net, may be insufficient. It may for example produce segmentations with a very thin rim in a subject where a thick and a thin rim are equally likely. The full distribution of possible segmentations $p(\mathbf{s}|\mathbf{x})$ matching a given image $\mathbf{x}$ contains valuable information for the downstream classification task. In recent years, several techniques have been proposed which allow to approximate this conditional probability distribution. In this work, we compare four such techniques: The probabilistic U-Net (Kohl et al., 2018), PHiSeg (Baumgartner et al., 2019), MC Dropout (Kendall et al., 2016), and ensembles (Lakshminarayanan et al., 2017).

The **probabilistic U-Net** (Kohl et al., 2018) is a combination of the conditional VAE (Sohn et al., 2015) approach with a U-Net architecture. **PHiSeg** further extends the idea by a hierarchical latent space and was shown to provide closer approximations of $p(\mathbf{s}|\mathbf{x})$. Both techniques estimate the *aleatoric* uncertainty.

**Ensembles** are implemented by training ten standard U-Nets with different random seeds (Ensemble$_{seeds}$). We additionally create an ensemble by training a U-Net for each of the 11 expert annotators in our data (Ensemble$_{experts}$). The widely used **MC Dropout** technique produces probabilistic segmentation samples by repeatedly predicting segmentations for the same image with dropout enabled. We use a dropout rate of 0.2 on the activation maps for training and testing. Dropout is applied to all layers except the final four segmentation layers. Ensembles and MC Dropout estimate *epistemic* uncertainty. We refer the reader to (Abdar et al., 2021) for definitions of aleatoric and epistemic uncertainty.

We use the improved U-Net architecture first proposed in Kohl et al. (2018) for all approaches with the exception of PHiSeg. PHiSeg employs the same U-Net encoder, but requires a specific decoder. Crucially, all examined probabilistic segmentation methods allow the generation of segmentation samples $\mathbf{s}_i$ from the estimated distribution of $p(\mathbf{s}|\mathbf{x})$.

### 2.3. Rim Thickness Curve (RTC) Extraction

Next, we extract RTCs $\mathbf{r}_i$ from segmentation samples $s_i$. Rim thickness is defined as the width of the rim between the borders of the optic cup and disc (Spaeth et al., 2002; Hwang and Kim, 2012). To compute the RTC, a beam centered at the optic cup is rotated by 360 degrees. At every half-degree interval, the points where the beam intersects with the borders of the optic disc and optic cup are determined. The rim thickness is calculated as the Euclidean distance between these two intersections. This process results in a data vector of length 720 for each segmentation sample $s_i$. The resulting RTCs were visualized as polar plots (see Fig. 1 and 3). We denote the rim-thickness extraction procedure as a deterministic function $g : \mathbf{s}_i \mapsto \mathbf{r}_i$. A visual explanation is shown in Appendix C.

### 2.4. Glaucoma Classification

The RTCs $\mathbf{r}_i$ are classified as "glaucoma suspect" or "not glaucoma suspect". We use a logistic regression classifier since, in preliminary experiments, more powerful classifiers such as SVMs or Random Forests did not lead to improvements. To prevent overfitting, the RTC data is reduced by grouping the values into 72 bins and calculating their mean. We train an individual classifier for the RTCs obtained from each of the examined segmentation methods. The optimal decision threshold for each classifier is obtained by maximizing the Youden index (sensitivity + specificity - 1) on the validation set. This results in a deterministic classifier $f : \mathbf{r}_i \mapsto d_i$ that maps each rim thickness sample to a binary diagnosis.

### 2.5. Uncertainty Estimation and Robust Classification

The final probability $p(d|\mathbf{x})$ of an input image $\mathbf{x}$ being "glaucoma suspect" is obtained by marginalizing over all possible segmentations, that is $p(d|\mathbf{x}) = \int p(\mathbf{s}|\mathbf{x})f(g(\mathbf{s}))d\mathbf{s}$. We approximate this integral using the Monte Carlo method with samples from the respective segmentation techniques. We use 100 samples for all methods, except for ensembles which are limited by the number of networks trained.

The probability $p(d|\mathbf{x})$ is a natural measure for uncertainty as it can be interpreted as a respective agreement or disagreement of the predictions resulting from the segmentation samples. In that sense it is comparable to expert disagreement. We obtain the final robust prediction of our pipeline by thresholding the above probability at 0.5.

## 3. Experiments and Results

### 3.1. Data

We used two publicly available fundus image datasets for experiments. The **Chákṣu dataset** (Kumar et al., 2023) contains 1345 fundus images with cup and disc annotations by five experts for each image. Additionally, each expert also provided a diagnosis

Table 1: **Quantitative results.** AUROC, sensitivity and specificity refer to the glaucoma classification task. Dice between the mean predicted segmentation and the consensus ground truth segmentation. The correlation coefficient (CC) between the mean pipeline prediction and the mean expert prediction for all test samples.

| | Segm. source | AUROC | | Sensitivity | | Specificity | | Dice | CC |
|---|---|---|---|---|---|---|---|---|---|
| | | RTC | CDR | RTC | CDR | RTC | CDR | | |
| *Det.* | U-Net | 0.857 | 0.8881 | 0.701 | 0.771 | 0.838 | 0.845 | 0.919 | - |
| *Probabilistic* | Ensemble$_{experts}$ | 0.863 | **0.890** | 0.719 | 0.789 | 0.802 | 0.853 | 0.907 | 0.629 |
| | Ensemble$_{seeds}$ | **0.899** | 0.886 | 0.824 | 0.736 | 0.792 | **0.888** | **0.926** | 0.639 |
| | Prob. U-Net | 0.876 | 0.869 | **0.877** | 0.824 | 0.749 | 0.799 | 0.871 | 0.612 |
| | PHiSeg | 0.884 | 0.882 | 0.807 | **0.842** | 0.741 | 0.752 | 0.900 | **0.653** |
| | MC Dropout | 0.885 | 0.887 | 0.736 | 0.736 | **0.835** | 0.874 | 0.894 | 0.592 |
| | Expert Annotations | 0.957 | 0.930 | 0.921 | 0.883 | 0.884 | 0.866 | - | - |
| | ResNet50 (black box) | 0.884 | | 0.754 | | 0.853 | | - | - |

of "glaucoma suspect" or "not glaucoma suspect". The dataset also contains consensus segmentations obtained using the STAPLE algorithm (Warfield et al., 2004). The **RIGA dataset** (Almazroa et al., 2018) consists of 750 fundus images with cup and disc annotations by six experts for each image. In contrast to Chákṣu it contains no diagnosis labels, and was only used for training the segmentation networks in our study.

We split the training portion of the Chákṣu data into a training and validation set according to an 80/20 split. We used the official test split of the Chákṣu data for all our evaluations. Additionally, we split the RIGA dataset into a training, and a validation portion according to a 80/20 split.

### 3.2. Training

We trained the segmentation-based stages of our pipelines with combined RIGA and Chákṣu datasets (see Sections 2.1 & 2.2), and the classifiers using only the Chákṣu dataset which contains Glaucoma suspect labels (see Section 2.4). More details about the training and model selection can be found in Appendix A.

### 3.3. Findings

**Uncertainty quantification improves downstream predictions**

In order to show that accounting for the uncertainty in the segmentation step leads to improved performance in downstream tasks, we included a deterministic U-Net for the cup and disc segmentation as an additional baseline. Again, we used the improved architecture proposed in Kohl et al. (2018). Moreover, we included a classifier trained directly on the expert annotations, as well as a black box ResNet50 network trained on the ROIs of the Chákṣu dataset to get a sense of the maximum achievable performance.

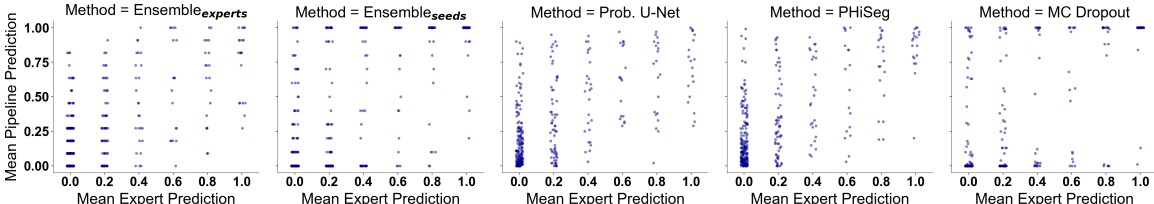

Figure 2: **Mean expert disagreement versus mean pipeline prediction for all methods.** Values close to 1 or 0 indicate large agreement for "glaucoma suspect" or "not glaucoma suspect", respectively. Values close to 0.5 denote high disagreement.

We observed that using probabilistic segmentation techniques consistently led to improvements in the downstream classification performance compared to the deterministic U-Net (Tab. 1), with Ensemble$_{seeds}$ achieving the highest overall AUC score.

Notably, the black box ResNet50 model performed slightly worse than the best probabilistic pipeline approaches. This suggests that our interpretable approach has the potential to outperform black-box models in certain settings. We note that black box approaches may perform better in a data-richer setting (De Vente et al., 2023).

Most methods showed similar performance in the segmentation task, as indicated by the Dice scores in Table 1. However, despite its high Dice score, the deterministic U-Net fell short on AUROC scores, indicating that accurate segmentation alone is insufficient, and that considering the entire probability distribution of the upstream task leads to improvements.

**RTCs outperformed CDR for glaucoma diagnosis**

In order to confirm the hypothesis that RTCs are better suited for glaucoma diagnosis than the widely used CDR, we additionally extracted the area CDR from all segmentations and trained an additional set of logistic regression classifiers on those values. We observed that the best AUROC scores were achieved with RTC, with particularly large improvements over CDR for the highly performing Ensemble$_{seeds}$.

**Propagated uncertainty correlates with expert disagreement**

We additionally calculated the Pearson's correlation coefficient (CC) between the mean expert prediction (i.e. all expert predictions averaged) and the mean pipeline prediction $p(d|\mathbf{x})$ in the last column of Tab. 1. PHiSeg achieved the highest CC with Ensemble$_{seeds}$ also performing very well. Visual inspection of the mean expert and pipeline predictions confirmed these findings (see Fig. 2). This indicates that the propagated uncertainty correlates with the expert disagreement, and thus is an informative measure for prediction uncertainty. However, further improvements may be achieved in future work by specifically optimizing downstream calibration.

**Qualitative analysis shows good segmentation and RTC agreement with experts**

The entropy maps and RTCs in Fig. 3 confirm that the distributions of the annotator disagreement approximately matched the estimated segmentation and RTC uncertainties.

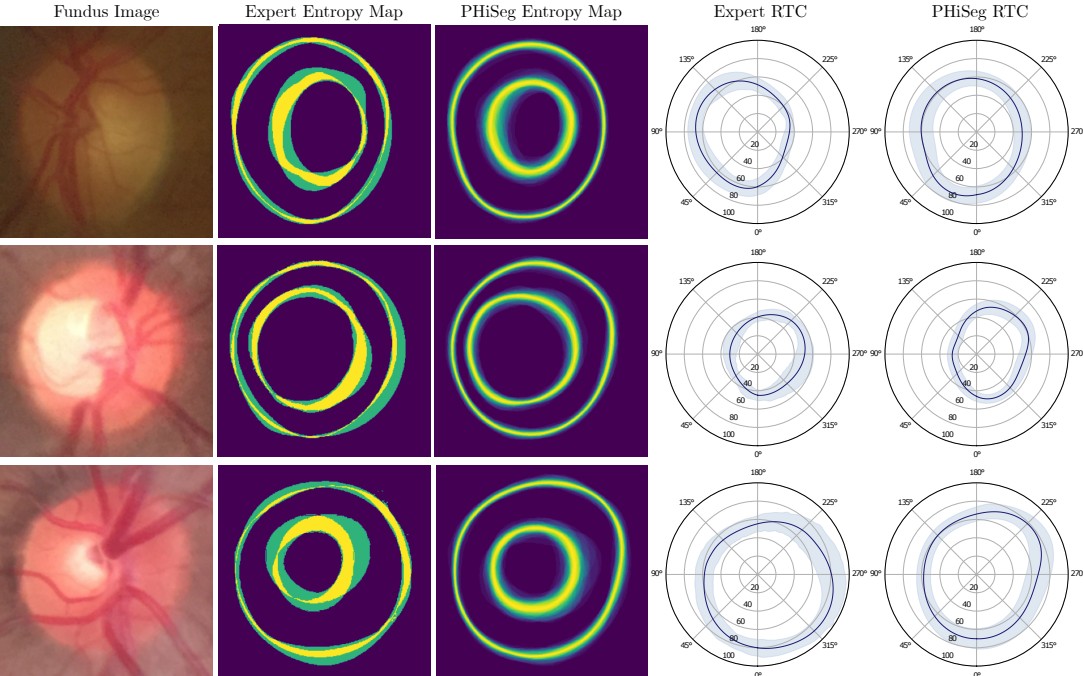

Figure 3: **Entropy maps and RTCs for expert annotations as well as PHiSeg.**
The mean rim-thickness (blue line) as well as the standard deviation (light blue shading) are displayed. Three different scenarios are shown: an uncertain case (top), a certain glaucoma suspect (middle) and a certain healthy eye (bottom).

We observed that the method correctly predicted higher uncertainties in areas where the rim was obscured by blood vessels. Additional samples and methods are shown in Appendix B.

## 4. Discussion and Conclusion

Here, we proposed a pipeline for human-interpretable glaucoma prediction. We showed that probabilistic segmentation techniques are suitable for capturing uncertainties in the location of the cup and disc, and demonstrated an approach for propagating these uncertainties through the pipeline steps to the final prediction. Knowledge about the uncertainty adds an additional level of interpretability to the individual pipeline steps. We furthermore proposed a simple strategy for obtaining robust predictions by marginalizing over the distribution of possible segmentations, and showed that accounting for the uncertainty in this manner led to improved downstream predictions. Our analysis of different probabilistic segmentation techniques revealed that a simple random seed ensemble provided the best balance. However, PHiSeg provided the best qualitative results and correlation with expert disagreements.

A limitation of our work is the sole focus on rim thickness as diagnostic feature. Future work will focus on incorporating diagnostic markers such as the color and intensity of the optic cup, and whether fundus corresponds to the right or left eye into our pipeline. These features are known to be diagnostically important and may further improve performance.

## Acknowledgments

This work was supported by the German Science Foundation (BE5601/8-1 and the Excellence Cluster 2064 "Machine Learning — New Perspectives for Science", project number 390727645) and the Hertie Foundation. The authors thank the International Max Planck Research School for Intelligent Systems (IMPRS-IS) for supporting Paul Fischer.

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

## Appendix A. Training Details

The ROI extraction U-Net network (see Sec. 2.1) was trained with the combined RIGA and Chákṣu training data sampling different expert ground truths for each batch. Model selection was performed using the Dice score on the validation set.

The probabilistic segmentation networks described in Sec. 2.2 were trained in the same fashion based on the regions extracted in the first step. Model selection for the probabilistic U-Net, PHiSeg, and MC Dropout was performed using the generalized energy distance (GED) metric (Kohl et al., 2018) between the expert annotations and the samples on the validation sets. The ensembles were analogously trained by sampling different experts for each batch, and model selection was performed based on the Dice score of the individual networks.

We trained classifiers with RTCs resulting from each of the segmentation methods as described in Sec. 2.4. The classifiers were trained using the Chákṣu training data, and the optimal threshold was determined using the Chákṣu validation data. We used 100 RTC curve samples per training image for all methods except the $\text{Ensemble}_{experts}$ and $\text{Ensemble}_{seeds}$, which were limited by design to 11 and 10 samples, respectively.

The ResNet50 baseline was initialized using ImageNet weights, and then fine-tuned on predicting the glaucoma label from the automatically extracted ROI crops of the Chákṣu dataset. During training we used random horizontal flips, and small random rotations in $[-10°, 10°]$ to augment the small dataset. Note that we did not use any augmentation for the segmentation networks, as they were additionally trained with RIGA data and segmentation tasks typically require fewer training data points due to the dense segmentation annotations. We used the AUC computed on the validation set for model selection. In order to compute the sensitivity and specificity in Table 1, we obtained the decision threshold which maximizes the Youden index.

## Appendix B. Additional qualitative results

In Fig. 4 we show additional qualitative results of the entropy for experts and models as well as the corresponding RTCs.

## Appendix C. Visual explanation of RTC extraction

Fig. 5 provides some intuition for the calculation of the RTC.

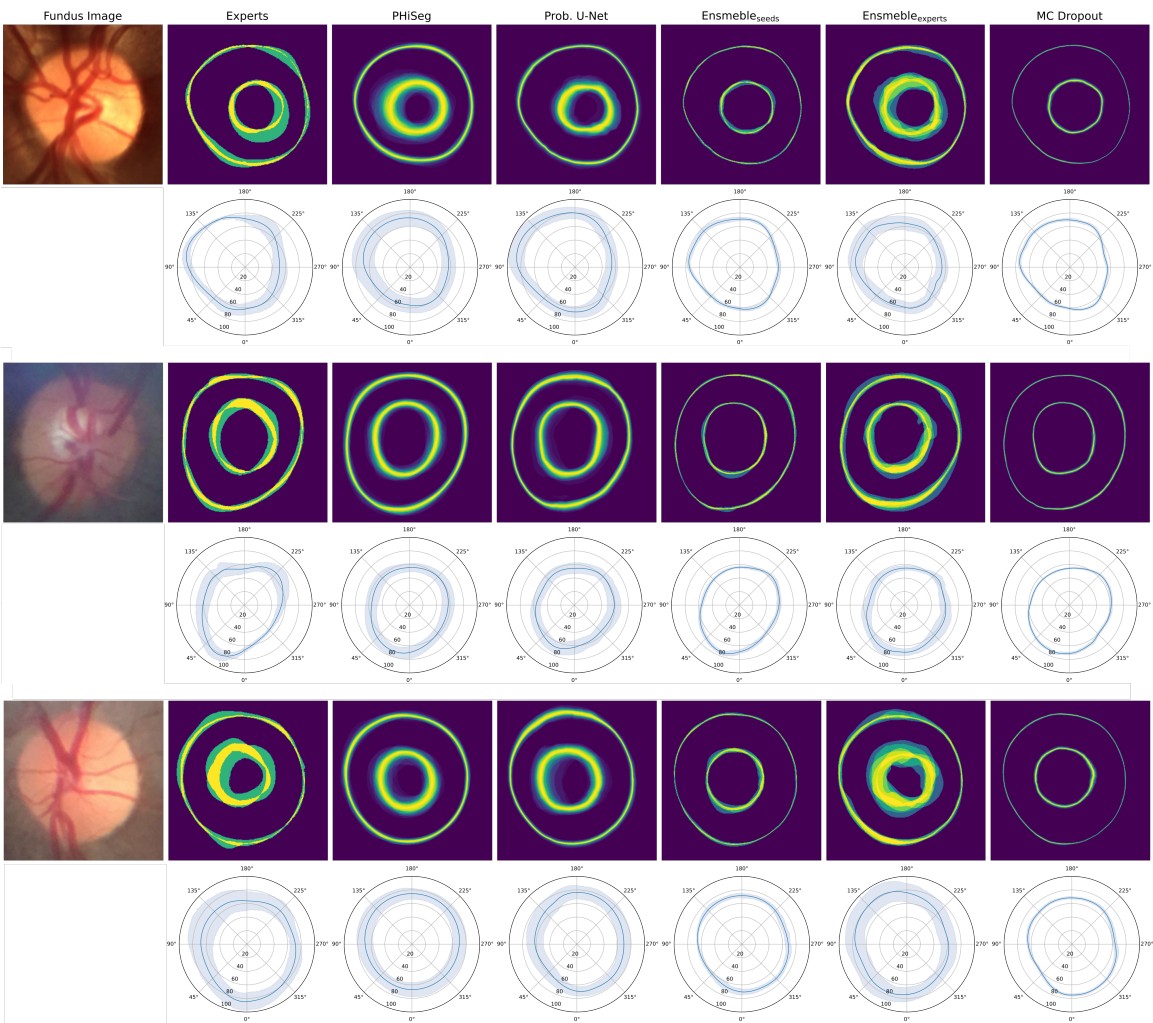

Figure 4: Entropy maps and RTC plots for every model for three representative example subjects.

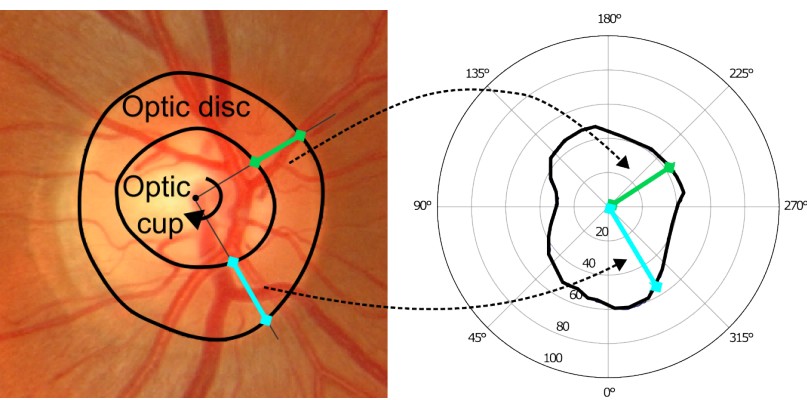

Figure 5: **Rim-thickness calculation.** A beam centered at the cup is rotated 360 degrees. At every 0.5 degrees the Euclidean distance between the cup and the disc is recorded in the rim thickness plot.

