# OpenReview forum: "Leveraging Probabilistic Segmentation Models for Improved Glaucoma Diagnosis: A Clinical Pipeline Approach"
_MIDL.io/2024/Conference — MIDL 2024 Poster_

### Official Review · Reviewer_btqo · 2024-02-28

**Confidence:** 3
**Preliminary Rating:** 4
**Final Rating:** 5

**Summary:**

The paper presents a multi-stage pipeline for enhancing glaucoma diagnosis through probabilistic segmentation of optic cup and disc in fundus images. It introduces an approach that leverages uncertainty quantification to improve the reliability of glaucoma detection, contrasting with traditional methods that may not account for inherent ambiguities. The authors demonstrate how the integration of probabilistic models and the extraction of Rim Thickness Curve (RTC) instead of the traditional Cup-to-Disc Ratio (CDR) can provide better diagnostic performance. They validate their approach with experiments on the Chaksu test set, showing improved classification accuracy and alignment with expert disagreement.

**Strengths:**

* This approach integrates the latest clinical recommendations with rim thickness curve (RTC) as an alternative to the traditional Cup-to-Disc Ratio (CDR).
* The paper introduces a novel probabilistic segmentation technique for the optic cup and disc in fundus images which is a critical step in glaucoma diagnosis.
* Quantification and utilization of uncertainty: a major strength is the paper's focus on quantifying and leveraging uncertainty in the segmentation process. By propagating this uncertainty through the pipeline and using it to enhance downstream performance, the paper addresses a crucial gap in current methodologies, which often inadequately handle uncertainty.
* Experiments and analysis: the paper’s experiments are exhaustive with a comparative analysis of four widely used probabilistic segmentation techniques and the validation of the proposed methods on two public datasets.
* The paper follows a clear and logical structure.

**Weaknesses:**

* It can be argued that little methodological improvement has been done from Fischer et al. work on uncertainty propagation.
* Reproducibility: code is not provided.
* Limitations discussion: The authors do not discuss the limitations of their approach.
* I suggest reviewing the grammar and typos of some sentences. I’ve provided a non exhaustive list in the detailed comments section.

**Detailed Comments:**

* Since you are using an improved version of the UNet architecture. Why did you choose the one provided by Kohl et al. and not nnUNet ?
* As you are providing a new baseline for glaucoma diagnosis, it would be great if you can also provide your code. This would help the community for future benchmarks based on RTC.
* Non exhaustive list of typos:
   - “with an comparative analysis” -> “with a comparative analysis”
   - “produces a predicted diagnoses” -> “produces a predicted diagnosis”
   - “sampling based” -> “sampling-based” (multiple occurrences)
   - “approach consisting of a ROI extraction” -> “consisting of an ROI”
   - “the aleotoric uncertainty” -> “the aleatoric uncertainty”
   - “100 samples for all methods, except the for ensembles” -> “except for ensembles”
   - “annotations to get a sense for the maximum ” -> “get a sense of the maximum”
   - “We observed that that using probabilistic” -> “observed that using probabilistic”
   - “We observed [...] consistently lead” -> “consistently led”
   - “While it produced sensible average behavior, individual segmentation samples exhibit”-> “exhibited”
   - “RTC agreement to experts” -> “RTC agreement with experts”

**Justification Of Final Rating:**

The authors have addressed my comments and clarified aspects of their paper on the rebuttal process. I would like to thank them for their efforts. The paper introduces a new pipeline that integrates uncertainty quantification into the segmentation of the optic cup and disc in fundus images for glaucoma diagnosis. The method is backed by a rigorous set of experiments and the contribution is significant. I suggest a strong accept.

**Justification Of The Preliminary Rating:**

The paper has a significant contribution, the method is novel and backed by a rigorous set of experiments. Results validate the authors’ contribution of a human-interpretable glaucoma prediction pipeline.

**Questions To Address In The Rebuttal:**

See detailed comments above. I suggest reviewing the grammar and spelling of the entire paper.

---

> ### Author Response · Authors · 2024-03-15
> **Rebuttal**
>
> We sincerely thank the reviewer for their time, valuable feedback, and positive comments. We summarized and re-structured the reviewer's points with the identifiers R4.x in order to facilitate any subsequent discussion.
>
> All changes to the revised PDF that we uploaded are highlighted in **orange**.
>
> > **R4.1:** It can be argued that little methodological improvement has been done from Fischer et al. work on uncertainty propagation.
>
> *Response:* While Fischer et al. have contributed towards uncertainty propagation, **this paper is building upon that work by further examining whether uncertainty propagation can *enhance* performance**. To this end, we propose a novel approach to marginalise over the distribution of the uncertain pipeline step, and show that this approach leads to superior performance compared to a deterministic method. **We have emphasized this more in the text**.
>
> > **R4.2:** Reproducibility: code is not provided.
>
> *Response:* The code was not yet ready for publication by the time of our initial submission. It is now available at https://github.com/annawundram/glaucoma-diagnosis-pipeline.
>
> > **R4.3:** Limitations discussion: The authors do not discuss the limitations of their approach.
>
> *Response:* **We have included a brief discussion of the limitations of our approach** in the Discussion.
>
> > **R4.4:** I suggest reviewing the grammar and typos of some sentences
>
> *Response:* We thank the reviewer for taking the time to put together the list of typos. **We fixed all of them in the updated PDF.** We have furthermore **thoroughly checked the revised manuscript for spelling and grammar mistakes**.
>
> > **R4.5:** Since you are using an improved version of the UNet architecture. Why did you choose the one provided by Kohl et al. and not nnUNet?
>
> *Response:* Our choice of architecture was motivated by our wish to use the probabilistic U-Net and PHiSeg as originally proposed by the respective authors. In order to make the results of the other techniques comparable to those methods, we chose this modified U-Net as our base architecture for all experiments. **We have clarified this in the updated PDF.**

---

### Official Review · Reviewer_fSYN · 2024-02-29

**Confidence:** 4
**Preliminary Rating:** 4
**Recommendation:** Poster
**Final Rating:** 5

**Summary:**

The authors present research on using deep learning in glaucoma diagnosis while incorporating it into physicians' workflow. The method takes into account uncertainty in the segmentation of eye components in the image. The method also employs a different metrics compared to commonly used ratio for diagnosis of the disease.

**Strengths:**

Overall, the project focuses on a clinically meaningful question of diagnosis of glaucoma, and takes an interesting angle on uncertainty analysis versus just segmentation. Different probablistic segmentation techniques are tried rigorously to examine results.
The authors showed clinically meaningful insights by comparing results of the probabilities from segmentation to clinical observations or clinician reviews.

**Weaknesses:**

Taking a step further and comparing propagated uncertainty, expert disagreement and actual diagnosis (may not be available) could be helpful.
It seems that propagated uncertainty from CDR as metric could also resemble expert disagreement.

**Detailed Comments:**

Overall I think this is a rigorous analysis focused on a specific and clinically meaningful question.

**Justification Of Final Rating:**

Thank you team for the update and detailed explanations for my questions. I find the added details convincing and suggest a strong accept for the conference. I encourage the authors to continue exploring this area.

**Justification Of The Preliminary Rating:**

Overall the focus area, method, and strength of results are acceptable for publication. There might be more consideration needed in alterantives and also in broadening scope, including details mentioned in the weakness/questions to address.

**Questions To Address In The Rebuttal:**

Taking a step further and comparing propagated uncertainty, expert disagreement and actual diagnosis (may not be available) could be helpful.
It seems that propagated uncertainty from CDR as metric could also resemble expert disagreement.

**Special Issue:**

No

---

> ### Author Response · Authors · 2024-03-15
> **Rebuttal**
>
> We sincerely thank the reviewer for their time, valuable feedback, and positive comments. We summarized and re-structured the reviewer's points with the identifiers R3.x in order to facilitate any subsequent discussion.
>
> All changes to the revised PDF that we uploaded are highlighted in **orange**.
>
> > **R3.1:** Taking a step further and comparing propagated uncertainty, expert disagreement and actual diagnosis (may not be available) could be helpful.
>
> *Response:* We thank the reviewer for this suggestion. Unfortunately, we haven't been able to find a dataset that contains multiple expert annotations **and** an actual (i.e. clinically confirmed) diagnosis. Unfortunately, the majority of datasets that we are aware of only contain labels such as "glaucoma suspect" or "referable glaucoma" as it is the case for example in Chaksu and AIROGS.
>
> > **R3.2:** It seems that propagated uncertainty from CDR as metric could also resemble expert disagreement.
>
> *Response:* We agree with the reviewer on this point. However, by using the rim thickness as a metric, the **CDR is also implicitly contained** as the CDR can be derived from the RTC.

---

> > ### Comment · Reviewer_fSYN · 2024-03-26
> > **Thank you**
> >
> > Thank you for your thoughtful responses to my questions and comments.

---

### Official Review · Reviewer_AME7 · 2024-03-04

**Confidence:** 5
**Preliminary Rating:** 4
**Recommendation:** Poster
**Final Rating:** 5

**Summary:**

The authors build a diagnostic pipeline for glaucoma from retinal images that puts to use a probabilstic unet-like system into a clinically-inspired multi-step process. Since the net can produce multiple samples from a posterior distribution of the OD and the cup, those are used to propagate such uncertainty into the measurement of a relevant biomarker for detecting glaucoma. Interestingly, the popular OD/cup ratio is not the measure of choice, but another one called rim-to-disc ratio, which seems to be more valid (kumar 2019)

**Strengths:**

- It is very nice to see papers that attempt to bridge clinical reality and technically interesting methods for which it is hard to see real applications (probabilistic unet and friends).
- The exploration of a recently proposed biomarker instead of the good old OD/Cup ratio.
- The datasets choice is meaningful.
- The method is exlpainable/transparent by design.

**Weaknesses:**

- It is okay to go from images to biomarkers and then use logistic regression to reach a diagnosis, as a way of increasing transparency. However, I believe this should be done without sacrificing (much of) the accuracy that a standard CNN would provide. Maybe the authors should spend a bit of time training an end-2-end CNN that takes OD crops and returns glaucoma diagnostic to check that the AUROC is not abismally higher.
- For glaucoma analysis, a more realistic setup could be a screening-like situation, with much more presence of the non-pathological class, and using other relevant metrics. I understand that extreme class imbalance does not make up for a nice situation, but maybe if we are to transition to real-life use-cases, then this should be the main assumption. Please see below for some suggestions :)
- Although using the rim-to-od ratio is a nice addition, I also find it a missing opportunity to verify that it is actually superior to the traditional od-to-cup ratio. The cited clinical literature supporting the benefits of the former is a 2002 ophthalmology paper (22 years ago this was proposed and it has not been widely picked up, one wonders why), and a NSR from the same authors of the Cháksu dataset, and this cannot be considered strong clinical evidence I think. It would have been great to see if using the conventional OD-to-cup ratio actually is worse through the eyes of the logistic regression by the end of the pipeline.

**Detailed Comments:**

I gave many details above, please see below for suggestions.

**Justification Of Final Rating:**

Not only my questions but those of other reviewers have been properly addressed. This is a solid paper that deserves to be accepted in my opinion. OpenReview still wants me to write fifty more chars...

**Justification Of The Preliminary Rating:**

I like this paper a lot, but I still would like to hear the authors' answers to my questions. Happy to increase my score if they manage to convince me. It seems I still need to write some more characters.

**Questions To Address In The Rebuttal:**

* I guess that it would be great if the authors could train a cnn to output glaucoma diagnostic from the same data (resnet50 pretrained on imagenet should do, and this would be a quick experiment I believe, given there are relatively few images for training a classifier)
* Also retrain the system to use OD-Cup ratio instead of OD-rim, although I am not sure if this makes lot of sense. I am hoping the authors will clarify if that is the case, or they can push back and tell me that it would be meaningless, no problem with that :)
* If the authors were interested in pursuing the "screening setup" direction, now or in the future, please check this out:
1) https://www.sciencedirect.com/science/article/pii/S2666914523000325
2) https://arxiv.org/abs/2302.01738

which also involves uncertainty but in the out-of-distribution side of things: train only with gradable images, then test with ungradable -diagnostically useless images:
- Related to this, the same organizers are running a challenge about explainable glaucoma in which they use a similarly large scale dataset. In a sizeable subdataset (6k) there are also explanations of diagnostics by means of multi-label annotatations, with possible clinical reasons for the given diagnostic. In addition, there are two raters involved, and if they disagree an expert gives their opinion to disambiguate diagnostic. Considering the interest of the authors on explainability and on retinal image-based disease diagnosis, I guess this and the above datasets must be mouth-watering ;)

**Special Issue:**

No

---

> ### Author Response · Authors · 2024-03-15
> **Rebuttal**
>
> We would like to sincerely thank the reviewer for their time, valuable feedback, and positive comments. We summarized and re-structured the reviewer's points with the identifiers R2.x in order to facilitate any subsequent discussion.
>
> All changes to the revised PDF that we uploaded are highlighted in **orange**.
>
> > **R2.1:** Maybe the authors should spend a bit of time training an end-2-end CNN that takes OD crops and returns glaucoma diagnostic to check that the AUROC is not abismally higher.
> >
> *Response:* We agree with the reviewer and **have therefore trained and evaluated a ResNet50 on the Chaksu optic disc crops** with the same train/val/test split as used in the paper (see new results in Table 1). The training details are reported in Appendix A. This end-to-end network reached an AUROC score of 0.884, which is slightly worse than the best probabilistic pipeline approaches. This shows that our probabilistic pipeline approach has the potential to perform on a par, or even outperform a black box approach. We acknowledge, however, that a black box model may have better performance in a data-richer setting as is for example suggested by the AIROGS challenge results.
>
> > **R2.2:** For glaucoma analysis, a more realistic setup could be a screening-like situation, with much more presence of the non-pathological class, and using other relevant metrics. (...)
>
> *Response:* As our paper aims at closing the gap to real life applications, **we agree that realistic class-imbalance is a necessary factor** to take into account when evaluating such pipelines. In our case, **we have worked with a class-imbalance of about 80/20 for non-pathological/pathological cases**. It can, however, be argued that this is not sufficient and real life scenarios might be even more imbalanced. We will further explore this in future work, potentially using the datasets the reviewer pointed out under R2.4.
>
> > **R2.3:** Although using the rim-to-od ratio is a nice addition, I also find it a missing opportunity to verify that it is actually superior to the traditional od-to-cup ratio.
>
> *Response:* If we understand correctly, then this point is asking for a comparison of RTC (rim thickness curve) to CDR (cup-to-disc ratio). We would like to emphasise that **our results include such a comparison** by drawing the reviewer’s attention towards **Table 1** which includes all metrics for both RTC and CDR as well as the section titled **“RTCs outperformed CDR for glaucoma diagnosis“ in Section 3.3.** If we have misunderstood the reviewer, we would kindly ask them to clarify their concern.
>
> > **R2.4:** If the authors were interested in pursuing the "screening setup" direction, now or in the future, please check this out (...)
>
> *Response:* We thank the reviewer for pointing us towards these interesting datasets. We were indeed unaware of the REGAIS dataset, and the many possibilities for future research it offers. We are very interested to pursue this in future work.

---

> ### Comment · Reviewer_AME7 · 2024-03-21
>
> Hello,
>
> Thanks for your answers, sorry to ask for something that you already had in the paper, my mistake for reading too fast, I think I am overworking these days. Anyway, I like the paper and the responses are satisfactory I believe. As a suggestion, it would be interesting to look into the samples where a black box prediction differs from the clinically-inspired system, and see why that is the case.
>
> By the way, next time a reviewer pushes to discuss about the difference between aleatoric and epistemic methods, and its impact, you may point them towards this 2024 ICLR oral (not mine, unfortunately):
>
> https://arxiv.org/abs/2401.08501
>
> where they show that for uncertainty in segmentation problems, basically in practice the epistemic vs aleatoric distinction is BS. Good luck!

---

> > ### Author Response · Authors · 2024-03-22
> >
> > We are glad to hear that the responses were satsifactory and helped to clarify your concerns! Thanks for the suggestion with the analysis of samples with differing diagnoses. That will be an interesting addition to a potential extension of this work.
> >
> > Thanks also for the paper on aleotoric/epistemic uncertainties. We had a hunch that the distinction might not be so clear-cut in practice for a while now. Great to see it confirmed!

---

### Official Review · Reviewer_Xv95 · 2024-03-04

**Confidence:** 3
**Preliminary Rating:** 3
**Final Rating:** 5

**Summary:**

In this paper, the authors introduce a new pipeline that integrates uncertainty quantification into the segmentation of the cup and disc in fundus images for diagnosing glaucoma. The proposed pipeline comprises four main stages: Initially, a U-Net model is trained to identify the regions of interest (ROI) by segmenting the cup and disc in the global image. Subsequently, a probabilistic model is applied to these ROIs to generate multiple segmentation samples of the cup and disc. For each segmentation proposal, the rim thickness curve (RTC) is computed and input into a linear classifier for diagnosis prediction. The final diagnostic score is derived by aggregating the predictions from all segmentations.

**Strengths:**

- The paper introduces well the context of the study, including the data, the medical problem at hand, and the objectives.
- The authors conducted extensive experiments across two different datasets, demonstrating that incorporating uncertainty significantly narrows the performance gap between deterministic models and expert-level diagnosis.
- The explanation and rationale behind each step of the pipeline are clear and well-articulated.
- To the best of my knowledge, this work is the first to demonstrate the superiority of Rim Thickness Curve (RTC) analysis over the conventional Cup-to-Disc Ratio (CDR) in this context.

**Weaknesses:**

- The literature review on the role of uncertainty in enhancing clinical diagnosis appears incomplete. The assertion that only a few studies have shown the benefits of integrating uncertainty might overlook significant contributions in the field, such as those focusing on probabilistic attention [1], and robustness in medical image classification[2]. It would also be beneficial to include references that clearly define "epistemic" and "aleatoric" uncertainty, to help the reader better understand these concepts that are not defined in the paper [3].
- The paper analysis of the results produced by the benchmarked methods lacks a discussion in terms of their focus on epistemic versus aleatoric uncertainty. An extended discussion in this context, potentially combined with an examination of the correlation between models and experts uncertainty, would enhance the paper's value. (e.g. Do "epistemic approaches" outperform "aleatoric ones", what could we conclude from this, ... ?)
- There is insufficient discussion regarding how individual components of the pipeline contribute to the overall performance improvement. Identifying whether the observed performance enhancements are due to the consideration of multiple diagnostic proposals per patient or solely due to the improved performance of the classifier would be insightful. A suggested approach would be to train the classifier directly on the ground truth segmentations, and then apply this trained classifier to the segmentations produced by the evaluated stochastic segmentation models. This would allow for a direct assessment of the segmentations' impact on performance (in particular when comparing ensembling vs other benchmarked methods as the classifier is not trained on the same number of samples), or alternatively, an ablation study could be conducted.
- The details provided about the experimental setup, such as the number of samples used to train the classifier, need to be more explicit (e.g., is the classifier trained on 100 samples per training image for all methods except ensembling ?).
- The paper does not adequately address the performance of the first ROI identification step and the rationale behind the choice of classifier. Additional details on these mentioned "preliminary experiments" could clarify these choices.
- The claim of interpretability is questionable, given that the proposed stochastic method might obscure which specific segmentation contributes to the final predictions. A discussion on the interpretability of stochastic models versus deterministic models could clarify this point.



[1]: Schmidt, A., Morales-Álvarez, P., & Molina, R. (2023). Probabilistic Attention Based on Gaussian Processes for Deep Multiple Instance Learning. IEEE Transactions on Neural Networks and Learning Systems.
[2]: Abdar, M., Salari, S., Qahremani, S., Lam, H. K., Karray, F., Hussain, S., ... & Nahavandi, S. (2023). UncertaintyFuseNet: robust uncertainty-aware hierarchical feature fusion model with ensemble Monte Carlo dropout for COVID-19 detection. Information Fusion, 90, 364-381.
[3]: Abdar, M., Pourpanah, F., Hussain, S., Rezazadegan, D., Liu, L., Ghavamzadeh, M., ... & Nahavandi, S. (2021). A review of uncertainty quantification in deep learning: Techniques, applications and challenges. Information fusion, 76, 243-297.

**Detailed Comments:**

- References should be organized in chronological order for consistency and ease of navigation ("(Rasheed et al., 2023; Tulsani et al., 2021; Sevastopolsky et al., 2019)", beginning of 4th paragraph in the Introduction)
- In Figure 1, the description should be corrected from "majority vote" to "average," as a majority vote would suggest that the final prediction is defined as the most prevalent class among the predictions while it is the average here.
- The phrase "may not show the whole story" is too informal for academic writing and should be rephrased.
- The term "aleotoric uncertainty" should be corrected to "aleatoric uncertainty".
- An open question worth exploring in future work is how the introduced variance affects the performance of downstream tasks. While this point is mentioned for one method, expanding this discussion to the other approaches would be beneficial.
- The availability of the code used in the experiments is not mentioned.

**Justification Of Final Rating:**

The authors provided detailed responses to my concerns and those of other reviewers. The explanations provided, especially concerning the interpretability of the pipeline, have significantly clarified many aspects and addressed several of my previous comments.

**Justification Of The Preliminary Rating:**

While the paper exhibits many strengths, it also presents several significant drawbacks, particularly regarding the impact of uncertainty on the performance at various stages of the proposed pipeline, and the discussion of the results. I believe the limitations outlined earlier can be addressed by the authors during the rebuttal period.

**Questions To Address In The Rebuttal:**

The authors may focus on addressing the concerns highlighted in the weaknesses section, especially the insufficient exploration of how each component within the pipeline individually contributes to the enhancement of overall performance, and a more comprehensive analysis of the results.

---

> ### Author Response · Authors · 2024-03-15
> **Rebuttal**
>
> We sincerely thank the reviewer for their time, valuable feedback, and positive comments. We summarized and re-structured the reviewer's points with the identifiers R1.x in order to facilitate any subsequent discussion.
>
> All changes to the revised PDF that we uploaded are highlighted in **orange**.
>
> > **R1.1:** The literature review on the role of uncertainty in enhancing clinical diagnosis appears incomplete. (...) include references that clearly define "epistemic" and "aleatoric" uncertainty(...)
>
> *Response:* We have broadened the scope of the literature review and **included the references** pointed out. Additionally, we **refer to the suggested paper for a definition of the terms for aleatoric and epistemic uncertainty**.
>
> > **R1.2:** The paper analysis of the results produced by the benchmarked methods lacks a discussion in terms of their focus on epistemic versus aleatoric uncertainty.
>
> *Response:* We compared the classification performance between methods for aleatoric and epistemic uncertainties and **did not observe systematic differences between the two types of models**. Even though we believe that further analysis of the different uncertainty types is an important and fruitful research avenue, we do not discuss this further in the paper due to lack of interesting insights and space constraints.
>
> > **R1.3:** There is insufficient discussion regarding how individual components of the pipeline contribute to the overall performance improvement. (...) A suggested approach would be to train the classifier directly on the ground truth segmentations(...)
>
> *Response:* We thank the reviewer for the suggestion. We would like to note that **the purpose of the individual pipeline components was not to improve performance, but rather to improve interpretability by making the pipeline similar to clinical processes**. Nevertheless, it is an interesting question how replacing steps of the pipeline by black box components affects performance. As part of our response to Reviewer 2 (see R2.1) we have added a black box ResNet50 classifier trained directly on the ROIs to Table 1. As can be seen, **replacing the majority of the pipeline by a black box does not lead to improvements.** We believe that these results subsume the suggested ablation, as it is unlikely that replacing smaller subsets of the pipeline by black boxes would lead to larger improvements.
>
> > **R1.4:** The details provided about the experimental setup, such as the number of samples used to train the classifier, need to be more explicit (...)
>
> *Response:* We **added a more detailed explanation of the classifier training** (see Appendix A).
>
> > **R1.5:** The paper does not adequately address the performance of the first ROI identification step and the rationale behind the choice of classifier. (...)
>
> *Response:* ROI detection is a comparatively easy step, and we have never observed the ROI detection to fail in our experiments. Note that **a failure in the ROI detection would have been reflected in the Dice scores, and the final classification score**. For the classifier, we tested several popular classification methods (Logistic Regression, SVM, Random Forests). **Due to space limitations, we included only the best performing classification method (i.e. Logistic Regression)**.
>
> > **R1.6:** The claim of interpretability is questionable, given that the proposed stochastic method might obscure which specific segmentation contributes to the final predictions.(...)
>
> *Response:* We would like to clarify that **our claim for interpretability results from setting up the diagnosis framework as a *pipeline* that mirrors clinical practice** as opposed to a black box model. We would like to further emphasize that currently all samples contribute equally to the final prediction. We don't believe there is any sacrifice in interpretability compared to non-stochastic approaches. Both, the pipeline approach as well as the ability to visualise the prediction variance at each step of the pipeline make the model more interpretable than standard black box approaches. **We have further clarified this in the Discussion**.
>
> > **R1.7:** In Figure 1, the description should be corrected from "majority vote" to "average".
>
> *Response:* For binary classification, computing the average and picking the majority vote is **equivalent**. We chose the term **“majority vote” for consistency with the expert majority vote**.
>
> > **R1.8:** An open question (...) is how the introduced variance affects the performance of downstream tasks.
>
> *Response:* We thank the reviewer for this interesting suggestion for future work. We plan to address this question in a potential extension.
>
> > **R1.9:** Availability of the code
>
> *Response:* The code is now available at https://github.com/annawundram/glaucoma-diagnosis-pipeline.
>
> > **R1.10:** Minor points regarding wording and ordering of references.
>
> *Response:* **We incorporated these points** in our updated draft.

---

> > ### Comment · Reviewer_Xv95 · 2024-03-21
> >
> > Thank you for the detailed responses to my remarks and for clearly outlining the modifications made in the revised manuscript.
> >
> > All my concerns have been satisfactorily addressed, but I would like to add an additional comment:
> >
> > **R1.3**:
> > My remark was rather referring to the comparison between the different probabilistic segmentation techniques. I think a little ablation study using the same logistic regression classifier, trained on expert annotations, for all the probalistic segmentation approaches could have provided clearer insights into which strategy yields the highest quality samples (as the performance of the logistic regression classifier could be biased by the number of samples used for training for instance). However, following your clarification, I understand that this analysis may exceed the intended scope of your study.

---

> > > ### Author Response · Authors · 2024-03-22
> > >
> > > We are glad to hear that the responses were satsifactory and helped to clarify your concerns. We apologise for this misunderstanding. We have indeed experimented with using the same classifier trained on GT segmentations for all segmentation methods during the initial development of our method. Back then we actively decided against including this ablation, since we believe that the maximum achievable performance with each segmentation network is more informative than the performance with a fixed classifier. In the end, this is also what one would use in practice. We found that the performance with classifiers specific to the segmentation network was marginally higher than a classifier trained on the segmentation networks' outputs. However, the overall picture is similar with both evaluation strategies.

---

### Meta-Review · Area_Chair_F9Zk · 2024-04-03

**Recommendation:** Accept (Poster)
**Confidence:** 5

**Metareview:**

All reviewers unanimously agree that the authors have effectively addressed my feedback and provided clarity on various aspects of their paper during the rebuttal phase. The paper presents a novel pipeline that incorporates uncertainty quantification into optic cup and disc segmentation in fundus images for glaucoma diagnosis. The approach is supported by thorough experimentation, and its contribution is noteworthy. Therefore, I highly recommend its acceptance.

---

### Decision · Program_Chairs · 2024-04-06

Accept (Poster)